# Clinical Characteristics and Risk Factors of Tuberculosis in Children and Adolescents in Xinjiang, China: A Retrospective Analysis

**DOI:** 10.3390/tropicalmed10100293

**Published:** 2025-10-16

**Authors:** Tao Xin, Gaofeng Sun, Jiangbutaer Entemake, Beiming Zhang, Weiwei Jiao, Qifeng Li

**Affiliations:** 1Department of Pediatrics, The Infectious Disease Hospital of Xinjiang Uygur Autonomous Region, The Sixth People’s Hospital of Xinjiang Uygur Autonomous Region, Urumqi 830049, China; xintao0506163@163.com; 2Department of Science and Education, Xinjiang Institute of Pediatrics, Xinjiang Hospital of Beijing Children’s Hospital, Children’s Hospital of Xinjiang Uygur Autonomous Region, The Seventh People’s Hospital of Xinjiang Uygur Autonomous Region, Urumqi 830054, China; sungaofeng2022@163.com (G.S.); 17690921803@163.com (J.E.); kbdogezhang@163.com (B.Z.); 3Beijing Key Laboratory of Core Technologies for the Prevention and Treatment of Emerging Infectious Diseases in Children, Key Laboratory of Major Diseases in Children, Ministry of Education, National Clinical Research Center for Respiratory Diseases, Laboratory of Respiratory Diseases, Beijing Pediatric Research Institute, Beijing Children’s Hospital, Capital Medical University, National Center for Children’s Health, Beijing 100045, China

**Keywords:** tuberculosis, children and adolescents, clinical characteristics, risk factors, China

## Abstract

**Background:** Tuberculosis (TB) remains a major public health challenge among children and adolescents in high-burden countries. Xinjiang, the region with the highest TB incidence in China, has limited data on the clinical and epidemiological characteristics of pediatric TB. **Methods:** We conducted a retrospective cross-sectional study of children and adolescents (≤17 years) hospitalized with TB at a regional referral hospital in Xinjiang between 1 January 2020 and 31 December 2022. Demographic, clinical, and laboratory data were analyzed, and risk factors for extrapulmonary TB (EPTB) and severe TB were assessed. **Results:** A total of 253 patients were included, of whom 54.9% (139/253) had pulmonary TB (PTB) and 45.1% (114/253) had EPTB. EPTB was more common among children <5 years (78.9%, 15/19). The predominant clinical symptoms were fever (55.7%, 141/253), cough (66.8%, 169/253), fatigue (60.9%, 154/253), and night sweats (51.8%, 131/253). Tuberculous meningitis (TBM) was the most frequent EPTB manifestation (40.4%, 46/114). Younger age, rural residence, and absence of BCG vaccination were associated with a higher risk of EPTB. Laboratory findings showed high positivity rates for tuberculin skin test (96.1%, 99/103) and interferon-γ release assay (84.5%, 196/232), but low yields for smear microscopy and Xpert MTB/RIF, especially in EPTB cases. **Conclusions:** Pediatric TB in Xinjiang is characterized by a high burden of EPTB, particularly TBM in young children. Strengthening early diagnosis and improving access to effective diagnostic tools are essential to reduce morbidity and improve outcomes in this vulnerable population.

## 1. Introduction

Tuberculosis (TB), caused by *Mycobacterium tuberculosis* (MTB), remains a major global health threat, profoundly affecting the health and development of millions of children and adolescents. In 2023, an estimated 10.8 million new TB cases occurred worldwide, with 1.25 million deaths, making TB the leading cause of death from an infectious disease. Children and adolescents under 15 years accounted for about 12% of all cases [1]. Yet, TB in this age group is frequently underdiagnosed or underreported, largely due to limited diagnostic capacity, inadequate staffing, and a lack of expertise in pediatric TB [2,3]. In addition, TB in children often presents with nonspecific symptoms that can be mistaken for other diseases, further delaying diagnosis and treatment [4,5].

China remains one of the world’s high-burden countries, contributing approximately 6.8% of global TB cases. Within China, the Xinjiang Uygur Autonomous Region has consistently reported the highest incidence of pulmonary TB (PTB). Data from the Chinese Tuberculosis Information Management System between 2009 and 2018 showed an average PTB incidence of 155 per 100,000 population in Xinjiang, substantially higher than the national average [6]. Despite this, information on the clinical and epidemiological characteristics of TB in children and adolescents in this region is scarce. Importantly, epidemiological profiles vary significantly across settings: for example, pediatric TB in some African regions is often complicated by high HIV co-infection rates, a scenario uncommon in China [7]. Moreover, studies conducted at national referral centers, which draw patients from across the country, may not accurately reflect the regional burden or local risk factors [8].

To address this gap, we conducted a retrospective cross-sectional study of children and adolescents hospitalized with TB in a regional referral hospital for infectious diseases in Xinjiang. The objective was to describe the demographic, clinical, and laboratory features of pediatric TB in this high-burden setting, and to identify risk factors associated with severe disease forms, particularly extrapulmonary TB (EPTB).

## 2. Methods

### 2.1. Study Population

This retrospective cross-sectional study was conducted at the Infectious Disease Hospital of the Xinjiang Uygur Autonomous Region between 1 January 2020 and 31 December 2022. The study population included children and adolescents aged ≤17 years who were diagnosed with TB and admitted to the pediatric TB wards. For patients with multiple hospitalizations, only data from the first admission were analyzed.

Demographic and clinical data were extracted from medical records, and a standardized clinical information survey was completed for each patient (Appendix A). Variables collected included age, sex, place of residence, clinical manifestations, Bacillus Calmette–Guérin (BCG) vaccination status, contact history, time from symptom onset to hospital visit, and TB diagnosis.

The study was approved by the Ethics Committee of the Infectious Disease Hospital of Xinjiang Uygur Autonomous Region (No. 2024-018) and conducted in accordance with the Declaration of Helsinki. All data were anonymized and de-identified to ensure patient confidentiality, in compliance with national and institutional guidelines for human research.

### 2.2. Diagnosis of TB

PTB diagnosis followed the Chinese Pulmonary Diagnosis Criteria (WS288-2017) [9] and was classified into three categories: (1) confirmed case: positive result from smear microscopy, culture, molecular methods, or histopathological examination; (2) clinically diagnosed case: radiographic evidence consistent with TB plus at least one of the following: TB symptoms, positive tuberculin skin test (TST) or interferon-γ release assay (IGRA), histopathological findings, or bronchoscopy evidence; (3) non-TB case: alternative diagnosis established, with clinical resolution without anti-TB treatment.

EPTB was diagnosed according to the Chinese Classification of Tuberculosis (WS 196-2017) [10] and current clinical guidelines [11]. A “confirmed” case required compatible clinical and imaging features plus either (i) positive microbiology (smear, culture, or nucleic acid test) from affected tissue/fluid or (ii) characteristic histopathology showing granulomatous inflammation with caseation. When microbiologic or histologic proof could not be obtained, a “clinical diagnosis” of EPTB was acceptable after exclusion of alternative diseases and documented improvement on anti-tuberculosis therapy.

### 2.3. Laboratory Tests

Laboratory assessments included TST, IGRA, smear microscopy, culture, and molecular testing (Xpert MTB/RIF). Specimens for pathogen detection included sputum (induced when necessary), bronchial lavage fluid, and gastric aspirates for PTB cases, while tissue biopsies were used for EPTB cases.

A TST was considered positive if the mean induration diameter was ≥5 mm. IGRA (T-SPOT.TB) was considered positive when antigen-stimulated spot counts exceeded the nil-control by ≥6, or were at least double the nil value when background counts were 6–10.

### 2.4. TB Classification

TB was classified according to Chinese guidelines WS196-2017 [10]:

PTB: involvement of lungs, trachea, bronchi, pleura, or intrathoracic lymph nodes;

EPTB: involvement of organs or tissues outside the lungs (e.g., lymph nodes, abdomen, genitourinary tract, skin, bones, joints, or meninges);

Severe TB: Disseminated or uncontrolled disease, or TB causing compression or infiltration of adjacent neurological, vascular, bronchial, cardiac, or skeletal structures, resulting in functional impairment [12].

### 2.5. Statistical Analysis

Descriptive statistics were used to summarize categorical variables as frequencies and percentages. Subgroup comparisons were performed by sex, age, residence, contact history, and time from symptom onset to hospital visit. Patients were stratified into three age groups based on WHO criteria: <5 years, 5–<10 years, and 10–17 years. Time to hospital visit was categorized as ≤15, 16–30, 31–60, and >60 days.

Risk factors for EPTB and severe TB were analyzed using modified Poisson regression to calculate adjusted relative risks (RRs) in multivariable models. The chi-square test or Fisher’s exact test was used to compare clinical symptoms and laboratory test results between groups. All analyses were conducted using SPSS version 27.0, and a two-sided *p* value < 0.05 was considered statistically significant.

## 3. Results

### 3.1. Demographic and Clinical Data

A total of 253 children and adolescents diagnosed with TB were enrolled. As demonstrated in Table 1, 60.1% (152/253) of the participants were female. The median age was 12 years, with the age range of 7 months to 17 years. More than half of the patients (56.5%, 143/253) came from rural areas. Only 42.7% (108/253) had a documented BCG vaccination history, and 19.4% (49/253) reported household TB contact. PTB was diagnosed in 54.9% (139/253) of cases, while 45.1% (114/253) had EPTB, either alone or in combination with PTB. Tuberculous meningitis (TBM) was the most common form of EPTB (40.4%, 46/114), particularly in younger children. Severe TB accounted for 87.0% (220/253) of all cases.

### 3.2. Distribution of TB Types by Age

The distribution of TB types across different age groups was analyzed. As shown in Table 2, the 10–17-year-old age group was predominantly affected by PTB, with the proportion of EPTB cases decreasing with age. Further analysis of the EPTB subtypes revealed that TBM was the most prevalent among different age groups (Table 1).

### 3.3. Clinical Symptoms by TB Type

Fever, cough, fatigue, and night sweats were the most common clinical manifestations. As demonstrated in Figure 1, PTB cases were more likely to present with cough (84.2%, *p* < 0.001) and sputum production (70.5%, *p* < 0.001), while children with concurrent PTB and EPTB showed more systemic symptoms such as fatigue (78.0%, *p* = 0.005), appetite loss (59.3%, *p* = 0.011), and weight loss (57.6%, *p* < 0.001).

### 3.4. Risk Factors Associated with EPTB or Severe TB

As demonstrated in Table 3, risk factor analysis revealed that younger age, rural residence, and absence of BCG vaccination were significantly associated with EPTB. A higher prevalence of EPTB (78.9%) was observed in younger children (aged <5 years). The proportion demonstrated a decline with increasing age (5–<10 years: 63.6%, RR = 0.746, *p* = 0.047; 10–17 years: 33.9%, RR = 0.413, *p* < 0.001). Furthermore, children and adolescents residing in county and rural areas exhibited a heightened risk for EPTB (county: RR = 1.591, *p* = 0.042; rural: RR = 1.541, *p* = 0.043). Patients who had not received the BCG vaccination exhibited a significantly higher propensity to develop EPTB, with a rate of 77.8% compared to 41.7% in the vaccinated group (RR = 1.631, *p* = 0.007). Rural residence was also the only independent risk factor for severe TB (Table 4, RR = 1.147, *p* = 0.035).

### 3.5. Laboratory Results

As demonstrated in Table 5, laboratory testing demonstrated high positivity rates for TST and IGRA across both PTB and EPTB (TST: 98.4% for PTB, 92.3% for EPTB; IGRA: 84.5% for PTB, 85.6% for EPTB), but smear microscopy and Xpert MTB/RIF had much lower positivity rates, especially in EPTB patients (smear microscopy: 6.3% vs. 22.3%, *p* = 0.001; Xpert MTB/RIF: 31.4% vs. 53.1%, *p* = 0.004).

## 4. Discussion

This study provides the first comprehensive description of pediatric TB in Xinjiang, the region with the highest TB burden in China [6]. Our findings demonstrated that TB in children and adolescents in this area was characterized by a high prevalence of EPTB, particularly TBM, and by pronounced disparities related to age, sex, and rural residence. These results not only reflect the local epidemiological landscape but also resonate with broader global trends in childhood TB, offering valuable insights for prevention and control.

Children under five are known to be highly susceptible to TB [8,13], yet they comprised only 7.5% of cases in this study, mirroring Xinjiang’s pediatric surveillance (14.3% aged <5 vs. 71.7% aged 10–14) [14]. The gap reflects the non-specific presentation and low microbiological detection rate of TB in toddlers and the absence of systematic screening, whereas older children are caught by school-based programs [14]. Future control must therefore proactively seek TB in the under-fives: trace and test all household contacts of infectious adults, and always consider TB when “pneumonia” fails repeated antibiotic courses.

More than half of our cohort came from rural areas, underscoring the marked rural–urban disparities in TB burden. Similar inequalities have been documented across China and in other low- and middle-income countries (LMICs), where limited access to healthcare, poor living conditions, and high rates of malnutrition accelerate TB transmission and worsen outcomes [15,16,17,18]. A large genomic study in rural China confirmed that transmission intensity is substantially higher in remote areas compared with urban settings [15]. In sub-Saharan Africa, similar challenges contribute to the persistence of childhood TB despite expanded TB programs [19]. These findings highlight the urgent need for tailored interventions in rural communities, including improved diagnostic capacity, strengthened referral systems, and enhanced nutritional support.

The sex distribution in our cohort showed a predominance of females (60.1%), a finding that contrasts with the male-dominated burden described in many global reports [8,20,21]. Interestingly, recent studies in South Africa, Uganda, and India also observed increased susceptibility among adolescent girls [22,23,24]. Hormonal and nutritional factors may explain this divergence. Puberty-related hormonal shifts are known to influence host immunity, with estrogen promoting Th1-mediated responses while androgens suppress them [25]. In addition, adolescent girls in several settings, including Xinjiang, have been reported to have lower vitamin D levels and higher prevalence of anemia compared with boys [26,27]. These vulnerabilities may increase the risk of disease progression. Therefore, adolescent girls represent a subgroup that warrants particular attention in TB prevention strategies.

EPTB represented nearly half of all cases in this study (45.1%). This proportion is higher than that typically reported in adults (37.4%) [28], but aligns with pediatric studies from Beijing (54%) [8] and mainland China (45.76%) [29]. In contrast, studies from Syria and India reported much higher proportions, exceeding 70% [30,31]. The very high burden of EPTB in young children in our study (78.9% in those under five years) reflects their immature immune systems, which predispose them to disseminated infection. This age-dependent pattern has been consistently observed worldwide [8,32].

Among EPTB cases, TBM was the leading presentation (40.4%), similar to reports from Beijing (38.8%) and national data across China (34.2%) [8,29]. However, in some countries, such as Syria and India, lymphatic TB is more common [13,30]. The predominance of TBM in our cohort may reflect the referral pattern of severe cases to tertiary hospitals, as well as delayed diagnosis in rural areas. Clinically, this finding reinforces the importance of considering TBM in febrile children with systemic symptoms (such as fatigue, weight loss, appetite loss, etc.), particularly in high-burden settings, as delayed recognition often results in severe neurological sequelae.

Clinical manifestations in our cohort were broadly consistent with global observations, with fever, cough, and fatigue being most frequent. Systemic symptoms such as loss of appetite and weight loss were more common among patients with combined PTB and EPTB, reflecting the greater disease severity. These results align with studies from South Asia and Africa, where wasting syndromes are strongly associated with disseminated TB and poor outcomes [13,32]. Comprehensive management strategies should therefore integrate nutritional support and close monitoring of growth and weight.

Risk factor analysis identified young age, rural residence, and absence of BCG vaccination as significant predictors of EPTB. These associations are well supported by previous studies [8,33,34]. The protective effect of BCG vaccination against severe TB in children has been consistently demonstrated in both Chinese and international cohorts, including Argentina [33,34]. However, in our study, vaccination status was not documented for more than half of the patients, reflecting gaps in record-keeping that may hinder accurate assessment. Strengthening vaccination coverage and monitoring is therefore crucial.

Diagnostic challenges remain a major obstacle. While immunological tests (TST and IGRA) showed high positivity, their utility is limited by cross-reactivity and reduced specificity in high-burden settings [35]. Microbiological confirmation rates were low, particularly among EPTB patients, consistent with findings from multicenter studies in China and Africa [13,29]. The limited sensitivity of smear microscopy and culture is a persistent challenge, and although molecular assays such as Xpert MTB/RIF offer promise, their sensitivity remains suboptimal for extrapulmonary forms of TB. Expanding access to rapid molecular diagnostics, especially in rural clinics, and establishing specimen transport networks could substantially improve case detection.

## 5. Limitations

This study has several limitations that should be considered when interpreting the findings. First, its retrospective design relied on the accuracy and completeness of medical records, and variables such as BCG vaccination and contact history were partly missing and based on self-reports, potentially introducing information bias and misclassification that could compromise related analyses and risk estimates. Second, as the study was conducted in a single referral hospital in Xinjiang during the COVID-19 pandemic, potential under-representation of children who did not seek care and the unique regional and period-specific context limit the generalizability of the observed distributions and clinical characteristics to other settings or time frames. Finally, the lack of long-term follow-up prevented assessment of treatment outcomes or sequelae. Future multicenter, prospective studies with more comprehensive data collection are needed to validate these findings and to further clarify risk factors for severe and extrapulmonary TB in children.

## 6. Conclusions

Our findings illustrate both local vulnerabilities and global commonalities in pediatric TB. The high burden of EPTB and TBM in Xinjiang mirrors trends observed in other high-burden regions, while the female predominance and pronounced rural disparities point to unique local dynamics. Strengthening early diagnosis, ensuring universal BCG vaccination, and improving rural healthcare infrastructure are critical steps to reduce the pediatric TB burden. These strategies, coupled with integration of nutritional support and wider access to molecular diagnostics, will be essential to improve outcomes for children and adolescents in Xinjiang and similar settings worldwide.

## Figures and Tables

**Figure 1 tropicalmed-10-00293-f001:**
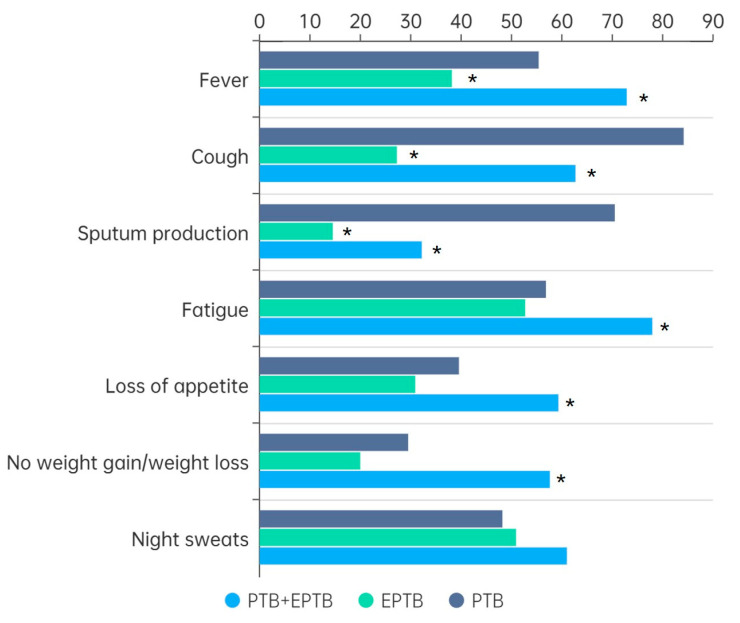
Comparison of clinical symptoms in patients with different types of TB. Note: PTB: pulmonary tuberculosis; EPTB: extrapulmonary tuberculosis; *: comparison with PTB group, *p* < 0.05.

**Table 1 tropicalmed-10-00293-t001:** The demographic and clinical characteristics of pediatric patients with TB.

Characteristics	*n* (%)
Sex	
Male	101 (39.9)
Female	152 (60.1)
Median age, y	12 (IQR: 8–14)
Age group, y	
<5	19 (7.5)
5–<10	66 (26.1)
10–17	168 (66.4)
Residence	
City	58 (22.9)
County	52 (20.6)
Rural	143 (56.5)
BCG vaccination	
Yes	108 (42.7)
No	9 (3.6)
Unknown	136 (53.7)
Time from onset of symptoms to hospital visit, d	
≤15	91 (36.0)
16–30	48 (19.0)
31–60	35 (13.8)
>60	79 (31.2)
TB type	
PTB	139 (54.9)
EPTB	55 (21.7)
PTB + EPTB	59 (23.3)
EPTB subtype	
Tuberculous meningitis	46/114 (40.4)
Osteoarticular TB	28/114 (24.6)
Lymphatic TB	27/114 (23.7)
Abdominal TB	6/114 (5.3)
Intestinal TB	6/114 (5.3)
Contact history	
Yes	49 (19.4)
No	93 (36.7)
Unknown	111 (43.9)
Severity of TB	
Severe	220 (87.0)
Non-severe	33 (13.0)
Clinical manifestations	
Fever	141 (55.7)
Cough	169 (66.8)
Sputum production	125 (49.4)
Fatigue	154 (60.9)
Loss of appetite	116 (45.8)
No weight gain or loss	86 (34.0)
Night sweats	131 (51.8)

Note: IQR: interquartile range; TB: tuberculosis; PTB: pulmonary tuberculosis; EPTB: extrapulmonary tuberculosis.

**Table 2 tropicalmed-10-00293-t002:** Classification of tuberculosis (TB) types stratified by age group.

	<5 Years N = 19, *n* (%)	5–<10 Years N = 66, *n* (%)	10–17 Years N = 168, *n* (%)
TB type			
PTB	4 (21.1)	24 (36.4)	111 (66.1)
EPTB	8 (42.1)	25 (37.9)	22 (13.1)
PTB + EPTB	7 (36.8)	17 (25.8)	35 (20.8)
EPTB subtype			
Tuberculous meningitis	6 (31.6)	20 (30.3)	20 (11.9)
Osteoarticular TB	2 (10.5)	11 (16.7)	15 (8.9)
Lymphatic TB	6 (31.6)	9 (13.6)	12 (7.1)
Abdominal TB	1 (5.3)	0 (0)	5 (3.0)
Intestinal TB	1 (5.3)	2 (3.0)	3 (1.8)
Others	0 (0)	2 (3.0)	7 (4.2)

**Table 3 tropicalmed-10-00293-t003:** Risk factors for extrapulmonary tuberculosis (EPTB) in this study.

Characteristics	Total (N = 253)	EPTB * (N = 114)	PTB (N = 139)	Adjusted RR (95% CI)	*p* Value
Age groups, y					
<5	19 (7.5)	15 (78.9)	4 (21.1)	1.00	Ref.
5–<10	66 (26.1)	42 (63.6)	24 (36.4)	0.746 (0.558–0.996)	0.047
10–17	168 (66.4)	57 (33.9)	111 (66.1)	0.413 (0.301–0.566)	<0.001
Sex					
Male	101 (39.9)	44 (43.6)	57 (56.4)	1.00	Ref.
Female	152 (60.1)	70 (46.1)	82 (53.9)	1.112 (0.847–1.460)	0.445
Residence					
City	58 (22.9)	18 (31.0)	40 (69.0)	1.00	Ref.
County	52 (20.6)	27 (51.9)	25 (48.1)	1.591 (1.018–2.488)	0.042
Rural	143 (56.5)	69 (48.3)	74 (51.7)	1.541 (1.014–2.341)	0.043
BCG vaccinated					
Yes	108 (42.7)	45 (41.7)	63 (58.3)	1.00	Ref.
No	9 (3.6)	7 (77.8)	2 (22.2)	1.631 (1.141–2.332)	0.007
Unknown	136 (53.7)	62 (45.6)	74 (54.4)	1.093 (0.812–1.471)	0.558
Time from onset of symptoms to hospital visit, d					
≤15	91 (36.0)	40 (44.0)	51 (56.0)	1.00	Ref.
16–30	48 (19.0)	21 (43.8)	27 (56.3)	1.053 (0.725–1.529)	0.787
31–60	35 (13.8)	15 (42.9)	20 (57.1)	1.005 (0.655–1.541)	0.982
>60	79 (31.2)	38 (48.1)	41 (51.9)	1.091 (0.801–1.486)	0.579
Contact history					
Yes	49 (19.4)	23 (46.9)	26 (53.1)	1.00	Ref.
No	93 (36.7)	42 (45.2)	51 (54.8)	1.034 (0.719–1.489)	0.856
Unknown	111 (43.9)	49 (44.1)	62 (55.9)	1.068 (0.727–1.568)	0.739

Note: PTB: pulmonary tuberculosis; EPTB: extrapulmonary tuberculosis; BCG: Bacillus Calmette–Guérin; RR: relative risk; *: the EPTB group encompasses both EPTB patients and patients with both EPTB and PTB.

**Table 4 tropicalmed-10-00293-t004:** Risk factors for severe tuberculosis in this study.

Characteristics	Total (N = 253)	Severe TB (N = 220)	Non-Severe TB (N = 33)	Adjusted RR (95% CI)	*p* Value
Age groups, y					
<5	19 (7.5)	14 (73.7)	5 (26.3)	1.00	Ref.
5–<10	66 (26.1)	57 (86.4)	9 (13.6)	1.171 (0.907–1.512)	0.227
10–17	168 (66.4)	149 (88.7)	14 (73.7)	1.210 (0.943–1.554)	0.135
Gender					
Male	101 (39.9)	88 (87.1)	13 (12.9)	1.00	Ref.
Female	152 (60.1)	132 (86.8)	20 (13.2)	1.002 (0.908–1.107)	0.963
Residence					
City	58 (22.9)	47 (81.0)	11 (19.0)	1.00	Ref.
County	52 (20.6)	40 (76.9)	12 (23.1)	0.958 (0.791–1.161)	0.665
Rural	143 (56.5)	133 (93.0)	10 (7.0)	1.147 (1.010–1.302)	0.035
BCG vaccinated					
Yes	108 (42.7)	91 (84.3)	17 (15.7)	1.00	Ref.
No	9 (3.6)	8 (88.9)	1 (11.1)	1.040 (0.828–1.306)	0.737
Unknown	136 (53.7)	121 (89.0)	15 (11.0)	1.066 (0.973–1.168)	0.169
Time from onset of symptoms to hospital visit, d					
≤15	91 (36.0)	80 (87.9)	11 (12.1)	1.00	Ref.
16–30	48 (19.0)	38 (79.2)	10 (20.8)	0.885 (0.750–1.044)	0.148
31–60	35 (13.8)	31 (88.6)	4 (11.4)	1.000 (0.876–1.142)	0.998
>60	79 (31.2)	71 (89.9)	8 (10.1)	1.023 (0.921–1.136)	0.671
Contact history					
Yes	49 (19.4)	42 (85.7)	7 (14.3)	1.00	Ref.
No	93 (36.7)	83 (89.2)	10 (10.8)	1.004 (0.885–1.140)	0.946
Unknown	111 (43.9)	95 (85.6)	16 (14.4)	0.934 (0.817–1.067)	0.315

Note: TB: tuberculosis; RR: relative risk; BCG: Bacillus Calmette–Guérin.

**Table 5 tropicalmed-10-00293-t005:** Laboratory test results for PTB and EPTB cases.

Diagnostic Test	Total (N = 253)	PTB (N = 139)	EPTB (N = 114)	*p* Value
TST (positive)	99/103 (96.1%)	63/64 (98.4%)	36/39 (92.3%)	0.151
IGRA (positive)	196/232 (84.5%)	107/128 (83.6%)	89/104 (85.6%)	0.678
Smear microscopy (positive)	35/225 (15.6%)	29/130 (22.3%)	6/95 (6.3%)	0.001
MTB culture (positive)	59/176 (33.5%)	38/100 (38%)	21/76 (27.6%)	0.149
Xpert MTB/RIF (positive)	82/183 (44.8%)	60/113 (53.1%)	22/70 (31.4%)	0.004

Note: PTB: pulmonary tuberculosis; EPTB: extrapulmonary tuberculosis; IGRA: interferon-γ release assay; TST: positive tuberculin skin test; MTB: *Mycobacterium tuberculosis.*

## Data Availability

The datasets generated and or analyzed during the current study are available from the corresponding author upon reasonable request.

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
