# Peer review of "Clinical Characteristics and Risk Factors of Tuberculosis in Children and Adolescents in Xinjiang, China: A Retrospective Analysis"

_tropicalmed, 2025, doi:10.3390/tropicalmed10100293_

Round 1

Reviewer 1 Report

Comments and Suggestions for Authors

Review: Clinical Characteristics and Risk Factors of Tuberculosis in Children and Adolescents in Xinjiang, China: A Retrospective Analysis.

This is a retrospective study of hospitalised children and adolescents with TB from a province in China from 2020-2022. Although of some interest, the study and manuscript have many shortcomings.

Major comments:

  1. General comments:
    - The authors use definitions that are not comparable to other studies and creating their own definitions is not helpful.
    A) Please consider changing age groups to same age groups as reporting to WHO: 0 to <5 years (not including 5 years of age); 5 to <10 years; 10-<15 years and 15-<19 years – The last two groups could possibly be combined as only adolescents up to 17 years were included.
    B) Dividing patients into pulmonary (PTB) and extrapulmonary TB (EPTB) based on the presence of EPTB is unusual – the usual way would be that PTB is the primary diagnosis and EPTB are the cases that do not have evidence of EPTB. The authors know (it seems) who had both PTB and EPTB therefore a better way to present the data would be to divide patients into three groups: PTB, EPTB and PTB plus EPTB. They can then still look at risk factors for EPTB by combining the last two groups, but it should clearly be stated/remembered that these hospitalised patients are a very biased group.
    C) It is not at all clear where the authors placed intrathoracic lymph nodes – are these placed with PTB or EPTB? I think that most paediatric TB experts would group them as part of pulmonary TB, but the authors need to clearly state this.
    D) The authors made up their own classification of severe TB and, per implication, non-severe TB. The WHO in the “WHO operational handbook on tuberculosis. Module 5: management of tuberculosis in children and adolescents. Geneva: World Health Organization; 2022. Licence: CC BY-NC-SA 3.0 IGO” provides a good classification for severe vs non-severe TB in children up to 16 years of age based on the trial published in NEJM by Turkova et al.”Shorter treatment for nonsevere tuberculosis in African and Indian children. N Engl J Med. 2022 Mar 10;386(10):911-922”. If this is not what they intend, then there is another classification published in PIDJ by Wiseman et al which will be very helpful in defining disease severity especially also including EPTB: “A proposed comprehensive classification of tuberculosis disease severity in children. Pediatr Infect Dis J 2012;31(4):347-52.”
  2. E) Disseminated TB – again, definition not standard – often used for miliary TB (not even mentioned once in the manuscript?) and central nervous system TB (e.g., TB meningitis). The authors include “TB in two non-contiguous sites”, which may be relevant, but what do they mean by this? Is PTB with cervical nodes “disseminated” or PTB and abdominal TB? I think they need to look at other published paediatric articles and clearly define what they mean and why it is important in their study.
  3. Introduction:
    - Lines 45-46: This sentence makes no sense - it seems as 10.8 million died of TB which is not correct - suggest reconstruction of this sentence - Estimated number of TB cases in 2023 was 10.8 million, with TB being the most common cause of death due to an infectious disease with XXX deaths in 2023.

- Lines 47-50: How does the authors explain this statement? Children are also misdiagnosed in high-income countries with low TB incidence, not only in low-income countries? I suggest delete first sentence and just say that TB in children is often misdiagnosed or not reported.
- Lines 55-58: This is an incorrect assumption – development of resistance even with incorrect treatment (what is “unregulated” treatment) is very rare in children (may not be irrelevant in adolescents) because of paucibacillary nature of TB in most children.
- Lines 68-69: What do the authors mean with this statement – not clear

  1. Methods:
    - Lines 92-100: This diagnostic definition could be fine for PTB, but where does microbiologically negative EPTB fit into this definition, as “radiographic” evidence of “TB” is needed? Please rethink these definitions clearly and also explain what is meant by “histopathological” confirmation of TB?
    - Lines 102-106: Need more information on tests, interpretation of test (e.g., what is defined as a positive tuberculin skin test); correct naming of microbiological tests (e.g. Xpert MTB/RIF or MTB/RIF Ultra?); what is “lavage fluid” and how does it differ from gastric aspirate?
    - Statistical analyses: Time interval of first 30 days – suggest divide into 0-15 and >15 to 30 days as well.
  2. Results:
    - The results should be drastically shortened by deleting all duplication in the text that is already mentioned in the tables. Refrain from discussing results in the results section – plainly give results.
    - The reviewer thinks that the results should be completely revisited in light of the General comments above
  3. Discussion:

- The authors should realise that with retrospective incomplete data (e.g. on BCG vaccination) and biased data (hospitalised “sicker” patients only) they should be very careful in their interpretation of the data. In the reviewer’s opinion the discussion is far too long with too many assumptions not based on results and it should be completely rewritten (also following points made in General comments) and much shorter.

Minor comments:
- Introduction: Once words are abbreviated, continue to use the abbreviation, not again the full word(s)
- Introduction: “prevalence” and “incidence” are used interchangeably, but these are not the same – please check and use correct terms
- Methods: Provide dates not only months of study?
- Methods, line 118 (and rest of manuscript) – not “gender” but “sex” – sex is biological and gender is by choice, so in children it should be sex   

Comments on the Quality of English Language

Construction of sentences in English should be changed so that the correct meaning of what the authors are trying to say is clear.

Author Response

Reviewer 1
- The authors use definitions that are not comparable to other studies and creating their own definitions is not helpful.
A) Please consider changing age groups to same age groups as reporting to WHO: 0 to <5 years (not including 5 years of age); 5 to <10 years; 10-<15 years and 15-<19 years – The last two groups could possibly be combined as only adolescents up to 17 years were included.

Reply: The age groups were reclassified as suggested. Please see page 3, line 113 and following analysis tables.

  1. B) Dividing patients into pulmonary (PTB) and extrapulmonary TB (EPTB) based on the presence of EPTB is unusual – the usual way would be that PTB is the primary diagnosis and EPTB are the cases that do not have evidence of EPTB. The authors know (it seems) who had both PTB and EPTB therefore a better way to present the data would be to divide patients into three groups: PTB, EPTB and PTB plus EPTB. They can then still look at risk factors for EPTB by combining the last two groups, but it should clearly be stated/remembered that these hospitalised patients are a very biased group.

Reply: The patients were divided into three groups as suggested. Please see Table 1 and Table 2.

  1. C) It is not at all clear where the authors placed intrathoracic lymph nodes – are these placed with PTB or EPTB? I think that most paediatric TB experts would group them as part of pulmonary TB, but the authors need to clearly state this.

Reply: The intrathoracic lymph nodes TB was in pulmonary TB group. Please see page 3, line 103.

  1. D) The authors made up their own classification of severe TB and, per implication, non-severe TB. The WHO in the “WHO operational handbook on tuberculosis. Module 5: management of tuberculosis in children and adolescents. Geneva: World Health Organization; 2022. Licence: CC BY-NC-SA 3.0 IGO” provides a good classification for severe vs non-severe TB in children up to 16 years of age based on the trial published in NEJM by Turkova et al.”Shorter treatment for nonsevere tuberculosis in African and Indian children. N Engl J Med. 2022 Mar 10;386(10):911-922”. If this is not what they intend, then there is another classification published in PIDJ by Wiseman et al which will be very helpful in defining disease severity especially also including EPTB: “A proposed comprehensive classification of tuberculosis disease severity in children. Pediatr Infect Dis J 2012;31(4):347-52.”

Reply: The severe TB was reclassified according to the definition in PIDJ article. Please see page 3, line 106-108.

  1. E) Disseminated TB – again, definition not standard – often used for miliary TB (not even mentioned once in the manuscript?) and central nervous system TB (e.g., TB meningitis). The authors include “TB in two non-contiguous sites”, which may be relevant, but what do they mean by this? Is PTB with cervical nodes “disseminated” or PTB and abdominal TB? I think they need to look at other published paediatric articles and clearly define what they mean and why it is important in their study.

Reply: Previously, disseminated TB was used to supplement the definition of severe TB. We have now adopted the literature-based definition of severe TB and consequently removed the definition of disseminated TB.

Introduction:
- Lines 45-46: This sentence makes no sense - it seems as 10.8 million died of TB which is not correct - suggest reconstruction of this sentence - Estimated number of TB cases in 2023 was 10.8 million, with TB being the most common cause of death due to an infectious disease with XXX deaths in 2023.

Reply: The sentence was rewritten as suggested. Please see page 1, line 43-44.

“In 2023, an estimated 10.8 million new TB cases occurred worldwide, with 1.25 million deaths, making TB the leading cause of death from an infectious disease.”

- Lines 47-50: How does the authors explain this statement? Children are also misdiagnosed in high-income countries with low TB incidence, not only in low-income countries? I suggest delete first sentence and just say that TB in children is often misdiagnosed or not reported.

Reply: The sentence was revised as suggested. Please see page 2, line 45-46.

“Yet TB in this age group is frequently underdiagnosed or underreported...”

- Lines 55-58: This is an incorrect assumption – development of resistance even with incorrect treatment (what is “unregulated” treatment) is very rare in children (may not be irrelevant in adolescents) because of paucibacillary nature of TB in most children.

Reply: The mentioned sentences were deleted as suggested. 

- Lines 68-69: What do the authors mean with this statement – not clear

Reply: The mentioned sentences were deleted to avoid confusion.

Methods:
- Lines 92-100: This diagnostic definition could be fine for PTB, but where does microbiologically negative EPTB fit into this definition, as “radiographic” evidence of “TB” is needed? Please rethink these definitions clearly and also explain what is meant by “histopathological” confirmation of TB?

Reply: The microbiologically negative EPTB was grouped into clinically diagnosed cases. The necessity for radiographic evidence (e.g. X-rays, ultrasounds, CT scans, MRIs, etc.) is paramount. Typical radiographic findings of EPTB include vertebral destruction, cold abscesses, and enlarged lymph nodes, amongst others. Histopathological findings refer to the detection of typical tuberculous granulomas with caseous necrosis in biopsy or post-operative specimens, with or without positive acid-fast staining or MTB nucleic acid detection.

- Lines 102-106: Need more information on tests, interpretation of test (e.g., what is defined as a positive tuberculin skin test); correct naming of microbiological tests (e.g. Xpert MTB/RIF or MTB/RIF Ultra?); what is “lavage fluid” and how does it differ from gastric aspirate?

Reply: The interpretation of tests was added as suggested. Please see page 3, line 97-100.

“A TST was considered positive if the mean induration diameter was ≥5 mm. IGRA (T-SPOT.TB) was considered positive when antigen-stimulated spot counts exceeded the nil-control by ≥6, or were at least double the nil value when background counts were 6–10.”

The name of microbiological test was revised as suggested. Please see page 3, line 94 and others through the manuscript.

The term “lavage fluid” refers to “bronchial lavage fluid”. It can be obtained during bronchoscopy procedures. While gastric aspirate is obtained from the stomach contents of patients when they are fasting.

- Statistical analyses: Time interval of first 30 days – suggest divide into 0-15 and >15 to 30 days as well.

Reply: The time interval was divided as suggested. Furthermore, the 61-90 and >90 days categories were combined to minimize the number of subgroups. Please see page 3, line 113-114.

Results:
- The results should be drastically shortened by deleting all duplication in the text that is already mentioned in the tables. Refrain from discussing results in the results section – plainly give results.

Reply: The results were shortened and revised as suggested. Please see page 3-8, line 121-180.

- The reviewer thinks that the results should be completely revisited in light of the General comments above

Reply: The results were revised according to the comments. Please see page 3-8, line 121-180.

Discussion:

- The authors should realise that with retrospective incomplete data (e.g. on BCG vaccination) and biased data (hospitalised “sicker” patients only) they should be very careful in their interpretation of the data. In the reviewer’s opinion the discussion is far too long with too many assumptions not based on results and it should be completely rewritten (also following points made in General comments) and much shorter.

Reply: The discussion was revised as suggested. Please see page 8-9, line 182-247.

Minor comments:
- Introduction: Once words are abbreviated, continue to use the abbreviation, not again the full word(s)

Reply: The full names and abbreviations were checked and revised as suggested.

- Introduction: “prevalence” and “incidence” are used interchangeably, but these are not the same – please check and use correct terms

Reply: The words “prevalence” and “incidence” are checked and revised as suggested. Please see page 1-2, line 41-66.

- Methods: Provide dates not only months of study?

Reply: The dates were added as suggested. Please see page 2, line 69-71.

“This retrospective cross-sectional study was conducted at the Infectious Disease Hospital of the Xinjiang Uygur Autonomous Region between January 1, 2020, and December 31, 2022.”

- Methods, line 118 (and rest of manuscript) – not “gender” but “sex” – sex is biological and gender is by choice, so in children it should be sex   

Reply: The word “gender” was replaced by “sex” as suggested. Please see the method part (page 2, line 76) and whole text.

Comments on the Quality of English Language

Construction of sentences in English should be changed so that the correct meaning of what the authors are trying to say is clear.

Reply: The language was carefully revised as suggested.

Reviewer 2 Report

Comments and Suggestions for Authors

Abstract:

Line 25: Retrospective analysis is not a study design. Was this an observational cross sectional, cohort or case control study? Was this a retrospective record review?

Line 26: how the age could be 0? Please amend throughout the manuscript.

Line 29: Severe TB?

Line 29: It is stated that patients with EPTB and severe TB were included, however, line 30 includes data about PTB and EPTB. Please amend.

Line 31: Age 0?

Line 34: Risk factors such as age and rural residence should have statistical values such as p-value, AOR, RR etc.

Line 39: It states that EPTB was common while line 31 states that EPTB was 45.1%. ???

Introduction

Line 49: This has resulted in a significant…….. This sentence does not co-relate/connect with previous sentence.

Line 50-58: This content should be summarized. Not related to scope of this manuscript.

Line 66: Is spatial clustering within the scope of this manuscript?

There is also a need to strengthen problem statement and rational of the study. Add more relevant studies.

Methods

Details about study design, data extraction form, and study variables are missing.

Line 70: Retrospective is in terms of secondary data. This is not the study design.

Line 112: Patients with PTB and EPTB were classified as EPTB? Is this correct or vice versa? What are the standard criteria?

Line 117-118: Sentence needs correction.

Line 118-122: Sentence should be modified/amended for clarity.

Results

Line 137: Delete.

Caption of Table 1 says that the data is about demographic characteristics of patients but it also includes clinical characteristics such as symptoms, TB type etc.

Line 140-154: It is highly recommended to present this data in Table. This will help better understanding of readers.

Line 157: Figure 1 is neither readable nor easily understandable. I would encourage the authors to carryout intersectional analysis and present in a Table.

Line 165-173: It is p value, not P value. It is not clear what statistical analysis was performed. Of course, PTB patients are more likely to have cough and sputum production compared with EPTB. How a patient with TB meningitis can produce sputum? In my opinion, these are not logical comparisons. The data presented in section 3.3 need a major re-write.

Line 176: A brief description of univariate and multivariate analysis should be presented. The description of significant findings should be supported with p value and RR. Further, the data in Table 1 does not include anything about server TB while text in line 176-183 has information about server TB. Please check this carefully.

Line 179: Age 0-5 is referent, while in Table 2, p value for adjusted RR is significant. Further, please see how the statement in line 178-179 justify the data in Table 2 (reference to PTB and EPTB).

Line 180-82: “Furthermore, children and adolescents residing in urban 180 areas exhibited a reduced likelihood of developing EPTB or severe TB in comparison to 181 those residing in rural or town areas” How this is supported by data in Table 2 (Check County, adjusted RR, PTB vs EPTB)

Table 3: p value is obtained by employing which statistical test?

Discussion

Discussion is too lengthy. It should focus on key findings and then discuss the impact of these findings on policy and practice. The discussion should ideally be not repetition of results. There is also a need to add references and findings of other local and global studies.

Suggested citations:

Treatment outcomes of childhood tuberculosis in three districts of Balochistan, Pakistan: findings from a retrospective cohort study. Journal of Tropical Pediatrics 67 (3), fmaa042

Comments on the Quality of English Language

The final manuscript should be read by a native English speaker for syntax and grammar. 

Author Response

Reviewer 2

Abstract:

Line 25: Retrospective analysis is not a study design. Was this an observational cross sectional, cohort or case control study? Was this a retrospective record review?

Reply: The study design is a retrospective cross-sectional study. Please see page 1, line 25.

“We conducted a retrospective cross-sectional study of children and adolescents….”

Line 26: how the age could be 0? Please amend throughout the manuscript.

Reply: The age was revised throughout the manuscript as suggested.

Line 29: Severe TB?

Line 29: It is stated that patients with EPTB and severe TB were included, however, line 30 includes data about PTB and EPTB. Please amend.

Reply: Line 29 is the method part. It means that we will explore the risk factors for EPTB and severe TB. Line 30 is the result part. It means that in this study, the patients can be categorized into PTB and EPTB subgroups, or severe TB and non-severe TB groups.

Line 31: Age 0?

Reply: The age was revised throughout the manuscript as suggested.

Line 34: Risk factors such as age and rural residence should have statistical values such as p-value, AOR, RR etc.

Reply: As age and residence were compared across three groups, multiple p-values and odds ratios were generated. Consequently, these values are not listed here.

Line 39: It states that EPTB was common while line 31 states that EPTB was 45.1%. ???

Reply: The sentence was revised as suggested. Please see page 1, line 36-37.

“Pediatric TB in Xinjiang is characterized by a high burden of EPTB…”

Introduction

Line 49: This has resulted in a significant…….. This sentence does not co-relate/connect with previous sentence.

Reply: The sentence is deleted as suggested.

Line 50-58: This content should be summarized. Not related to scope of this manuscript.

Reply: The content was shortened as suggested. Please see page 2, line 45-49.

“Yet TB in this age group is frequently underdiagnosed or underreported, largely due to limited diagnostic capacity, inadequate staffing, and a lack of expertise in pediatric TB [2,3]. In addition, TB in children often presents with nonspecific symptoms that can be mistaken for other diseases, further delaying diagnosis and treatment [4, 37].”

Line 66: Is spatial clustering within the scope of this manuscript?

Reply: The mentioned sentences were deleted to avoid confusion.

There is also a need to strengthen problem statement and rational of the study. Add more relevant studies.

Reply: More relevant studies were added as suggested. Please see page 2, line 56-61.

“Importantly, epidemiological profiles vary significantly across settings: for example, pediatric TB in some African regions is often complicated by high HIV co-infection rates, a scenario uncommon in China [7]. Moreover, studies conducted at national referral centers, which draw patients from across the country, may not accurately reflect the regional burden or local risk factors [10].”

Methods

Details about study design, data extraction form, and study variables are missing.

Reply: The missing information was added as suggested. Please see page 2, line 69-78. The data extraction form was uploaded as Table S1.

“This retrospective cross-sectional study was conducted at the Infectious Disease Hospital of the Xinjiang Uygur Autonomous Region between January 1, 2020, and December 31, 2022. The study population included children and adolescents aged ≤17 years who were diagnosed with TB and admitted to the pediatric TB wards. For patients with multiple hospitalizations, only data from the first admission were analyzed.

Demographic and clinical data were extracted from medical records, and a standardized clinical information survey was completed for each patient (Table S1). Variables collected included age, sex, place of residence, clinical manifestations, Bacillus Calmette-Guérin (BCG) vaccination status, contact history, time from symptom onset to hospital visit, and TB diagnosis.”

Line 70: Retrospective is in terms of secondary data. This is not the study design.

Reply: The study design was revised as suggested. Please see page 2, line 69.

“This retrospective cross-sectional study was conducted at the Infectious Disease Hospital…”

Line 112: Patients with PTB and EPTB were classified as EPTB? Is this correct or vice versa? What are the standard criteria?

Reply: The subgroups were revised as suggested. Now there are three subgroups: PTB, EPTB, PTB plus EPTB. Please see Table 1 and Table 2.

Line 117-118: Sentence needs correction.

Line 118-122: Sentence should be modified/amended for clarity.

Reply: The sentences were revised as suggested. Please see page 3, line 111-114.

“Subgroup comparisons were performed by sex, age, residence, contact history, and time from symptom onset to hospital visit. Patients were stratified into three age groups based on WHO criteria: <5 years, 5–<10 years, and 10–17 years. Time to hospital visit was categorized as ≤15, 16–30, 31–60, and >60 days.”

Results

Line 137: Delete.

Reply: The sentence was deleted as suggested.

Caption of Table 1 says that the data is about demographic characteristics of patients but it also includes clinical characteristics such as symptoms, TB type etc.

Reply: The caption of Table 1 was revised as suggested. Please see page 3, line 134.

“Table 1. The demographic and clinical characteristics of pediatric patients with TB.”

Line 140-154: It is highly recommended to present this data in Table. This will help better understanding of readers.

Reply: More information was combined into Table 1 and the paragraph was revised as suggested. Please see Table 1, and page 3, line 122-132.

“A total of 253 children and adolescents diagnosed with TB were enrolled. As demonstrated in Table 1, 60.1% (152/253) of the participants were female. The distribution by age and sex revealed a higher proportion of males in the <5 years age group (52.6%, 10/19), which decreased to 40.9% (27/66) in the 5-<10 years group and 38.1% (64/168) in the 10-17 years group. The median age was 12 years, with the age range of 7 months to 17 years. More than half of the patients (56.5%, 143/253) came from rural areas. Only 42.7% (108/253) had a documented BCG vaccination history, and 19.4% (49/253) reported household TB contact. PTB was diagnosed in 54.9% (139/253) of cases, while 45.1% (114/253) had EPTB, either alone or in combination with PTB. Tuberculous meningitis (TBM) was the most common form of EPTB (40.4%, 46/114), particularly in younger children. Severe TB accounted for 87.0% (220/253) of all cases.”

Line 157: Figure 1 is neither readable nor easily understandable. I would encourage the authors to carryout intersectional analysis and present in a Table.

Reply: Figure 1 was replaced by Table 2 as suggested. Please see page 5, line 143.

Line 165-173: It is p value, not P value. It is not clear what statistical analysis was performed. Of course, PTB patients are more likely to have cough and sputum production compared with EPTB. How a patient with TB meningitis can produce sputum? In my opinion, these are not logical comparisons. The data presented in section 3.3 need a major re-write.

Reply: The p value was revised as suggested. Please see Table 3-4 and others in the context.

The statistical analysis methods were added in the method part. Please see page 3, line 116-118.

“The chi-square test or Fisher’s exact test was used to compare clinical symptoms and laboratory test results between groups.”

The clinical symptoms of three TB subtypes (PTB, EPTB, PTB+EPTB) were compared to illustrate that the symptoms of PTB mainly restricted to the lung, PTB+EPTB cases are prone to develop systemic symptoms, while the symptoms of EPTB are the least typical. Please see page 5, line 146-150.

Line 176: A brief description of univariate and multivariate analysis should be presented. The description of significant findings should be supported with p value and RR. Further, the data in Table 1 does not include anything about server TB while text in line 176-183 has information about server TB. Please check this carefully.

Reply: The description was revised as suggested. Please see page 6, line 156-165.

“As demonstrated in Table 3, risk factor analysis revealed that younger age, rural residence, and absence of BCG vaccination were significantly associated with EPTB. A higher prevalence of EPTB (78.9%) was observed in younger children (aged <5 years). The proportion demonstrated a decline with increasing age (5-<10 years: 63.6%, RR=0.746, p=0.047; 10-17 years: 33.9%, RR= 0.413, p<0.001). Furthermore, children and adolescents residing in county and rural areas exhibited a heightened risk for EPTB (county: RR=1.591, p=0.042; rural: RR=1.541, p=0.043). Patients who had not received the BCG vaccination exhibited a significantly higher propensity to develop EPTB, with a rate of 77.8% compared to 41.7% in the vaccinated group (RR=1.631, p=0.007). Rural residence was also the only independent risk factor for severe TB (Table 4, RR= 1.147, p=0.035).”

Line 179: Age 0-5 is referent, while in Table 2, p value for adjusted RR is significant. Further, please see how the statement in line 178-179 justify the data in Table 2 (reference to PTB and EPTB).

Reply: I agree with you. The positions of the EPTB and PTB columns in Table 3 (previously Table 2) have been interchanged. Please see page 6, line 166.

Line 180-82: “Furthermore, children and adolescents residing in urban 180 areas exhibited a reduced likelihood of developing EPTB or severe TB in comparison to 181 those residing in rural or town areas” How this is supported by data in Table 2 (Check County, adjusted RR, PTB vs EPTB)

Reply: The positions of the EPTB and PTB columns in Table 3 (previously Table 2) have been interchanged. Then it is easy to find out that residence in rural area is risk factor for EPTB. Please see page 6, line 166.

Table 3: p value is obtained by employing which statistical test?

Reply: The statistical method was added in the method part. Please see page 3, line 116-118.

“The chi-square test or Fisher’s exact test was used to compare clinical symptoms and laboratory test results between groups.”

Discussion

Discussion is too lengthy. It should focus on key findings and then discuss the impact of these findings on policy and practice. The discussion should ideally be not repetition of results. There is also a need to add references and findings of other local and global studies.

Reply: The discussion was rewritten. Please see page 8-9, line 182-247.

Suggested citations:

Treatment outcomes of childhood tuberculosis in three districts of Balochistan, Pakistan: findings from a retrospective cohort study. Journal of Tropical Pediatrics 67 (3), fmaa042

Reply: The reference was added as suggested. Please see page 13, reference 36.

Comments on the Quality of English Language

The final manuscript should be read by a native English speaker for syntax and grammar. 

Reply: The language was carefully revised as suggested.

Reviewer 3 Report

Comments and Suggestions for Authors

The manuscript is well-written and addresses an important topic. Please consider expanding the limitations section, especially regarding the retrospective design and missing data (e.g., BCG vaccination, contact history). Strengthening the discussion with more comparison to regional and global pediatric TB studies would improve the manuscript.

Comments on the Quality of English Language

Overaly the English could be improved to more clearly express the research.

Author Response

Reviewer 3

Comments and Suggestions for Authors

The manuscript is well-written and addresses an important topic. Please consider expanding the limitations section, especially regarding the retrospective design and missing data (e.g., BCG vaccination, contact history). Strengthening the discussion with more comparison to regional and global pediatric TB studies would improve the manuscript.

Reply: The mentioned limitation was added as suggested. Please see page 9, line 251-252.

“Notably, data on BCG vaccination and contact history were frequently missing, limiting the robustness of related analyses.”

The discussion was rewritten. Please see page 8-9, line 182-247.

Comments on the Quality of English Language

Overaly the English could be improved to more clearly express the research.

Reply: The language was carefully revised as suggested.​

Round 2

Reviewer 1 Report

Comments and Suggestions for Authors

Review R1: Clinical Characteristics and Risk Factors of Tuberculosis in Children and Adolescents in Xinjiang, China: A Retrospective Analysis’

The manuscript is much improved but further corrections and clarifications are necessary. Comments as follow:

Major comments:

  1. Diagnostic criteria for diagnosing extrapulmonary TB (EPTB) are still not included – the “Diagnosis of TB” still focus on pulmonary TB – it should also include specifics on how EPTB was diagnosed, especially TB meningitis, which was the most common type of EPTB
  2. Abstract: Results section – percentages without numbers/denominators are very difficult to interpret, especially if the denominators used vary within the results section – these should be added
  3. References: Line 49 uses reference 37, which is definitely not in numerical order – please ensure that references are in correct order. The references themselves also need attention – the style should be consistent throughout and according to the journal’s guidelines for authors
  4. Methods – line 72: Were there children diagnosed with TB at this facility with non-severe TB that were discharged home rather than being admitted to the pediatric TB wards? This may create bias in your study and should be clarified.
  5. Results, line 124: The male/female ratio in the <5 year old children is not “higher” in males – it is the same as expected at this age.
  6. Table 1 – age group numbers: The authors should comment in the discussion on the very low number of children <5 years of age – this is expected to be much higher than the 5-<10-year-old age group in a setting with high TB burden. This is most likely because these younger children are missed (under-diagnosed) especially if there is a high reliance on positive tests for infection (as in this cohort) and/or on microbiological confirmation of TB
  7. Table 1 on BCG vaccination and further discussion/results regarding BCG vaccination. The authors interpret the “unknown BCG vaccination” as not receiving BCG vaccination, which is incorrect. They need to revise these statements and also make sure that differences between children BCG vaccinated and those with no/unknown BCG are statistically significant (which is not the case in Table 4. Please reconsider carefully (see for example line 162; line 232)
  8. Table 3 – last column not readable – needs to be corrected. Also provide percentages in column 1 (Total)
  9. Line 173 and following table is numbered incorrectly – this should be table 5
  10. Discussion, 1st paragraph. The reviewer thinks that the very high rate of TBM is due to selection bias of children with TB admitted to hospital (see comment 4 above) – this should be mentioned and also in the limitations.
  11. Line 200: The male-dominated burden is in adults – which is known – were these studies in adults or children/adolescents?
  12. Line 222: Diagnosing children with TBM only when they present with neurological signs is too late – what should clinicians be looking out for to diagnose these children earlier?

Author Response

Comment 1: Diagnostic criteria for diagnosing extrapulmonary TB (EPTB) are still not included – the “Diagnosis of TB” still focus on pulmonary TB – it should also include specifics on how EPTB was diagnosed, especially TB meningitis, which was the most common type of EPTB

Response 1: The diagnostic criteria for EPTB is added as suggested. Please see page 3, line 94-100.

“EPTB was diagnosed according to the Chinese Classification of Tuberculosis (WS 196-2017) and current clinical guidelines. A “confirmed” case required compatible clinical and imaging features plus either (i) positive microbiology (smear, culture, or nucleic-acid test) from affected tissue/fluid or (ii) characteristic histopathology showing granulomatous inflammation with caseation. When microbiologic or histologic proof could not be obtained, a “clinical diagnosis” of EPTB was acceptable after exclusion of alternative diseases and documented improvement on anti-tuberculosis therapy.”

Comment 2: Abstract: Results section – percentages without numbers/denominators are very difficult to interpret, especially if the denominators used vary within the results section – these should be added

Response 2: The numbers/denominators are added as suggested. Please see page 1, line 29-37.

Results: A total of 253 patients were included, of whom 54.9% (139/253) had pulmonary TB (PTB) and 45.1% (114/253) had EPTB. EPTB was more common among children <5 years (78.9%, 15/19). The predominant clinical symptoms were fever (55.7%, 141/253), cough (66.8%, 169/253), fatigue (60.9%, 154/253), and night sweats (51.8%, 131/253). Tuberculous meningitis (TBM) was the most frequent EPTB manifestation (40.4%, 46/114). Younger age, rural residence, and absence of BCG vaccination were associated with higher risk of EPTB. Laboratory findings showed high positivity rates for tuberculin skin test (96.1%, 99/103) and interferon-γ release assay (84.5%, 196/232), but low yields for smear microscopy and Xpert MTB/RIF, especially in EPTB cases.”

Comment 3: References: Line 49 uses reference 37, which is definitely not in numerical order – please ensure that references are in correct order. The references themselves also need attention – the style should be consistent throughout and according to the journal’s guidelines for authors

Response 3: The references order and style are revised as suggested.

Comment 4: Methods – line 72: Were there children diagnosed with TB at this facility with non-severe TB that were discharged home rather than being admitted to the pediatric TB wards? This may create bias in your study and should be clarified.

Response 4: Typically, following a diagnosis of tuberculosis in a child, the patient is admitted to hospital for intensive treatment, and only discharged once their condition has improved.

Comment 5: Results, line 124: The male/female ratio in the <5 year old children is not “higher” in males – it is the same as expected at this age.

Response 5: The difference is not statistically significant. So the sentence is deleted to avoid confusion.

Comment 6: Table 1 – age group numbers: The authors should comment in the discussion on the very low number of children <5 years of age – this is expected to be much higher than the 5-<10-year-old age group in a setting with high TB burden. This is most likely because these younger children are missed (under-diagnosed) especially if there is a high reliance on positive tests for infection (as in this cohort) and/or on microbiological confirmation of TB

Response 6: The discussion is added as suggested. Please see page 8, line195-202.

“Children under five are known to be highly susceptible to TB [10, 27], yet they comprised only 7.5% of cases in this study, mirroring Xinjiang’s pediatric surveillance (14.3% aged <5 vs 71.7% aged 10–14) [38]. The gap reflects the non-specific presentation and low microbiological detection rate of TB in toddlers and the absence of systematic screening, whereas older children are caught by school-based programs [38]. Future control must therefore proactively seek TB in the under-fives: trace and test all household contacts of infectious adults, and always consider TB when “pneumonia” fails repeated antibiotic courses.”

Comment 7: Table 1 on BCG vaccination and further discussion/results regarding BCG vaccination. The authors interpret the “unknown BCG vaccination” as not receiving BCG vaccination, which is incorrect. They need to revise these statements and also make sure that differences between children BCG vaccinated and those with no/unknown BCG are statistically significant (which is not the case in Table 4. Please reconsider carefully (see for example line 162; line 232)

Response 7: As demonstrated in Table 3, a total of 108 patients received BCG vaccination, of whom 41.7% subsequently developed EPTB. A total of nine patients did not receive the BCG vaccination, with 77.8% of these patients subsequently developing EPTB. The difference is found to be significant (RR=1.631, p=0.007), as previously outlined in the text. The patients with unknown BCG vaccination status were not discussed.

The Table 4 provides an analysis of the risk factors associated with severe TB. It is important to note that the only risk factor for severe TB is residence in a rural area. In the ensuing discourse, the focal point will be an examination of the risk factors associated with EPTB, not severe TB.

Comment 8: Table 3 – last column not readable – needs to be corrected. Also provide percentages in column 1 (Total)

Response 8: The Table 3 is revised as suggested. Please see page 6, line 172.

Comment 9: Line 173 and following table is numbered incorrectly – this should be table 5

Response 9: The Table number was corrected as suggested. Please see page 7, line 179 and page 8, line 184.

Comment 10: Discussion, 1st paragraph. The reviewer thinks that the very high rate of TBM is due to selection bias of children with TB admitted to hospital (see comment 4 above) – this should be mentioned and also in the limitations.

Response 10: As demonstrated in response to comment 4, there is no bias for the admission of TB patients. One potential explanation for the high rate of TBM observed in this study may be attributed to the fact that the study hospital functions as a TB referral hospital rather than a pediatric hospital. Please see page 9, line 233-235.

Comment 11: Line 200: The male-dominated burden is in adults – which is known – were these studies in adults or children/adolescents?

Response 11: The study population of reference 10, 16, 17 comprised children, adults and adolescents, respectively.

Comment 12: Line 222: Diagnosing children with TBM only when they present with neurological signs is too late – what should clinicians be looking out for to diagnose these children earlier?

Response 12: The discussion was revised as suggested. Please see page 9, line 235-237.

“Clinically, this finding reinforces the importance of considering TBM in febrile children with systemic symptoms (such as fatigue, weight loss, appetite loss, etc.)…”
